# Impact of HIV-1 Diversity on Its Sensitivity to Neutralization

**DOI:** 10.3390/vaccines7030074

**Published:** 2019-07-25

**Authors:** Karl Stefic, Mélanie Bouvin-Pley, Martine Braibant, Francis Barin

**Affiliations:** 1Inserm U1259, Université de Tours, 37000 Tours, France; 2CHRU de Tours, Centre National de Référence du VIH—Laboratoire Associé, 37000 Tours, France

**Keywords:** HIV-1 vaccine, broadly neutralizing antibodies, diversity, evolution, envelope

## Abstract

The HIV-1 pandemic remains a major burden on global public health and a vaccine to prevent HIV-1 infection is highly desirable but has not yet been developed. Among the many roadblocks to achieve this goal, the high antigenic diversity of the HIV-1 envelope protein (Env) is one of the most important and challenging to overcome. The recent development of broadly neutralizing antibodies has considerably improved our knowledge on Env structure and its interplay with neutralizing antibodies. This review aims at highlighting how the genetic diversity of HIV-1 thwarts current, and possibly future, vaccine developments. We will focus on the impact of HIV-1 Env diversification on the sensitivity to neutralizing antibodies and the repercussions of this continuous process at a population level.

## 1. Introduction

Prevention strategies against HIV-1 infection have been transformed in recent years with the implementation of the pre-exposure prophylaxis (PrEP) using antiretroviral drugs. The development of broadly neutralizing antibodies (bnAbs) that are now in clinical trials should provide additional possibilities for prevention [1,2,3,4,5,6]. Despite encouraging results of PrEP on HIV-1 incidence in some communities, challenges such has global drug availability, long term side effects of antiviral drugs, adherence to the prophylaxis methods, and social acceptability are limitations likely to persist. Therefore, efforts to develop an effective vaccine to stop the HIV/AIDS pandemic are still needed. The HIV-1 envelope glycoproteins, which mediate entry into host cells, are the target of neutralizing antibodies (NAbs) and constitute the key antigen for a prophylactic vaccine. So far, this armored door is locked and no antigenic form able to enter an efficient vaccine has been identified yet. This is largely due to the huge antigenic diversity of the envelope glycoproteins, which can represent up to 35% difference in amino acid sequences between subtypes. In this review, we will focus on the current knowledge on the HIV-1 envelope glycoprotein diversity and its evolving nature leading to a continuously increasing resistance to NAbs, with major consequences in the HIV-1 vaccine field.

## 2. HIV Env Diversity and Humoral Response

The HIV-1 envelope (Env) is the only target for NAbs on the surface of the virus. It is a trimer of gp41–gp120 heterodimers. As Env is the target of NAbs, the genetic and antigenic variability among HIV-1 remains a challenge to the development of an effective vaccine.

Cross-species transmission events from nonhuman primates to humans have generated four HIV-1 groups: M (major), O (outlier), N (non-M, non-O) and P. HIV-1 group M, responsible for the current pandemic, exhibits the most important viral diversity in the *env* gene [7,8,9,10], leading to 9 distinct subtypes (A–D, F–H, J, K), and numerous circulating recombinant forms (CRF) [7,10]. Globally, the subtype C predominates worldwide (representing 50% of HIV-1 viruses), followed by subtypes B and A, which account for about 10% of infections each. CRF02_AG, CRF01_AE, and subtype G are responsible for 5–8% of infections each and many other recombinant forms, which emerge regularly, generate diverse sub-epidemics [11]. Furthermore, viruses within the same clade differ by 8–17% (maximum 30%) in amino acid composition in the viral Env glycoprotein, whereas this difference is 17–35% between isolates from different subtypes, illustrating the outstanding variability of HIV-1 [7,8,9]. 

At the individual level, during the early months of HIV-1 infection, most of the patients develop autologous NAbs directed against the gp120 and gp41 subunits of the transmitted/founder (T/F) variant. The breadth of the neutralizing response is relatively narrow, as illustrated by its inability to neutralize heterologous isolates [12,13,14,15]. These antibodies do not seem to protect against disease progression but exert a selective pressure that drives the viral evolution and leads to the rapid selection of escape variants [16,17,18]. As a result, autologous NAbs are effective against variants of the viral quasi-species present several weeks or months earlier but appear to be unable to neutralize the contemporary variants [12,15,16]. The viral quasi-species varying in Env composition in each patient ultimately gives rise to a highly diverse virus population which coevolves in parallel with the antibody response [19]. 

The molecular basis of HIV-1 escape from autologous neutralization involves multiple diverse mechanisms. Autologous NAbs target mainly the surface-exposed regions, in particular the V1/V2 and V3 variable loops of gp120, which could explain their narrow specificity [20,21,22,23]. The general mechanisms leading to resistance to NAbs include single amino acid substitutions, insertions/deletions in the variable regions of Env, and an increased number and/or changes in positions of potential N-linked glycosylation sites (PNGS) at its surface (called glycan shield) [15,16,17,21,24,25]. The effect of these mechanisms on the emergence of antibody resistant variants within the viral quasi-species seems highly variable and heterogeneous according to the strains studied [17,21]. These modifications within the Env glycoproteins lead to an alteration of the target epitopes for NAbs, either by direct effects on the target epitope or by changes in the tertiary and quaternary structure of Env [26]. 

Nevertheless, 20–30% of HIV-1 infected individuals develop, after two or three years of infection, bnAbs able to block infection by diverse viruses from multiple subtypes [27,28,29]. Generally, this development correlates with high plasma viremia, increased viral diversity, and lower CD4+ T-cell counts [30,31,32]. The delayed bnAb response could be attributed to a slow antigen-dependent affinity maturation process and seems to result from exposure to an evolving antigen over many years [33,34,35]. Although bnAb responses fail to control viremia once infection has been established, a vaccine able to generate such responses prior to viral exposure could effectively prevent infection in humans, as the appropriate antibodies would be present before HIV diversification. 

## 3. BnAbs, Env Targets, and Cross-Subtype Neutralization

Among individuals developing bnAbs, a small fraction of patients (1%) called elite neutralizers, develop very broad and highly potent bnAbs able to neutralize isolates of various subtypes in vitro [27,36]. Since 2009, thanks to the development of single-cell-based antibody cloning techniques, a large number of bnAbs with outstanding breadth and potency have been generated from these elite neutralizers [37,38,39,40]. Passive transfer studies with bnAbs in nonhuman primates have demonstrated effective protection against simian/human immunodeficiency virus (SHIV) [41,42,43]. The bnAbs’ functionality lies in their ability to bind and clear both cell-free viruses and infected cells [44,45]. 

Characterizing the epitopes of these bnAbs on the HIV-1 Env trimers allow us to define the sites of vulnerability that should be targeted in a vaccine design [46,47,48]. These sites of vulnerability include the conserved regions near the CD4 binding site (CD4bs), the V1V2-glycan apex (V1V2g), the V3-glycan region (V3g), the gp41 membrane-proximal external region (MPER), and a complex region at the gp120-gp41 interface including the fusion peptide located at the N terminus of gp41 (Figure 1) [49,50]. 

The breadth and potency of bnAbs have been reviewed elsewhere [39]. The data inform us of the impact of the HIV-1 genetic diversity on the susceptibility to bnAbs. Transmitted-founder (T/F) viruses identified at the time of acute infection, or at least viruses isolated very early after primary infection, have specific properties that provide them a specific advantage for transmission [51,52,53]. Indeed, many studies have described a genetic bottleneck in the receiver at time of primary infection, suggesting that T/F viruses are those that must be preferentially targeted for protection. Thus, they are the most suited for evaluation of neutralization potency. 

Focusing on T/F viruses, the sensitivity to bnAbs seems associated with HIV-1 subtypes [54,55]. Extensive data are available for subtypes B and C, while those on non-B/non-C viruses are more limited. V1/V2g bnAbs are less potent against subtype B viruses [55,56], whereas bnAbs targeting the gp120-gp41 interface are more potent against this subtype [57,58]. The breadth of V3g bnAbs depends on the presence of a PNGS at position N332, whose frequency is variable both intrasubtype and intersubtype [59,60,61]. HIV-1 subtype B is more sensitive to CD4bs and MPER bnAbs than subtype C [60,62]. Therefore, the neutralization breadth of bnAbs depends on the subtype, as illustrated in Figure 2 for a selection of bnAbs against three panels of T/F viruses. Several factors may account for this variability. The efficacy of a particular bnAb against viruses from various subtypes may depend in part on the subtype infecting the individual from whom this bnAb was isolated [63]. The molecular patterns on the Env surface may differ between subtypes and also explain these differences [64,65,66,67,68]. As a consequence, extensive comparisons of Env structures across viral subtypes are necessary to identify signature sequences that might be associated with conserved sensitivity to neutralization in order to overcome viral diversity. Data have been generated by bioinformatic methods to trace the viral signatures associated with sensitivity to bnAbs [64,65,66,67,68]. They suggest that bnAbs with similar specificities show recurrent signature patterns [50,55,69,70,71]. Bricault et al. have defined signatures for four bnAbs classes (CD4bs, V1V2g, V3g, and MPER bnAbs) [55]. In a proof-of-concept exploration of signature-based epitope targeted (SET) vaccines, a trivalent vaccine with bnAb signatures defined for the V2 loop was employed to inform immunogen design in a guinea pig model. An increased neutralization breadth was observed, suggesting a key role of these Env molecular determinants [55]. Although the magnitude of the neutralization response remained low to moderate, these data can help guide the rational design of an effective HIV vaccine. 

## 4. Evolution of HIV-1 Env Towards a Greater Resistance to Neutralization over Time

Being transmitted from person to person, under constant—but different—selection pressure due to individual NAb repertoires in each individual, one might expect that HIV-1 Env evolves towards greater resistance to antibody neutralization not only within the host but also at the species level (Figure 3). 

Bunnik et al. suggested this hypothesis first and found pieces of evidence of this repercussion at the population level [72]. Comparing HIV-1 variants isolated shortly after seroconversion from patients of the Amsterdam Cohort either early in the epidemic (1985–1989) or later (2003–2006), they found an increasing resistance of subtype B viruses to NAbs during the course of the epidemic. We confirmed the phenomenon in another cohort of seroconvertors in France, among individuals infected by subtype B viruses at three periods of the epidemic, spanning more than 20 years (1987–2010) [62,73]. The same trend was observed using either sera from HIV-1 infected individuals or bnAbs [72,73]. This evolution was also observed more recently for T/F viruses of subtype C infecting individuals from several sub-Saharan African countries [60] and for T/F viruses of clade CRF02_AG infecting individuals in France [61]. Through the analysis of *env* sequences, these studies showed that the continued diversification of HIV-1 over time was associated with increasing resistance to NAbs (Figure 4). Concordant observations in different locations, at different periods of time, and for different genotypes allows us to conclude that there is a drift towards increasing resistance to neutralization of HIV-1 at the population level, which may reflect an adaptation of HIV-1 to the human species. This finding may have major consequences for vaccine development. In addition, the potency of the neutralizing response measured in HIV-1 infected individuals tends to decrease over time [72,73], possibly due to the evolution of HIV-1 to become less immunogenic, though confounding factors might exist. Still, there is other evidence of HIV-1 adaptation at the population level, both an adaptation to human immune cellular responses [74] and an evolution towards greater virulence [75,76,77]. 

The protection efficiency conferred by a prophylactic vaccine is thought to rely in large part on its ability to trigger the production of bnAbs [78,79]. Therefore, studying the sensitivity to bnAbs of HIV-1 over time would be an interesting proxy to better characterize the evolution towards resistance, quantitatively and qualitatively. A thorough investigation using some of the most potent bnAbs against subtype B viruses found increasing resistance to bnAbs targeting all major sites of vulnerability (V1V2g, V3g, CD4bs, and the gp41-gp120 interface), except gp41 MPER [62]. On the bright side, this reflects a strong natural selection pressure and the humoral response against these four targets, which might be triggered by a putative vaccine. However, this also shows that HIV-1 is able to escape from all these bnAbs without losing its capacity to infect new individuals over time. The constant sensitivity to bnAbs targeting the gp41 MPER could be due to a lack of selective pressure or to higher functional constraints in this region [62,80,81]. Interestingly, different profiles of evolution were observed for other genotypes. Subtype C viruses have become more resistant to VRC01 (CD4bs), PG9 (V3g), and 4E10 (MPER) but not to CAP256-VRC26.25 (V1V2g) or PGT128 (V3g) [60]. For CRF02_AG the increasing resistance was found significant only for bnAbs targeting the CD4bs, although the study period might have been too short to get complete information for other specificities [61]. In the course of the continuous drift of HIV-1 genome and epitopes, it is possible that different clades follow separate evolution pathways regarding resistance to neutralization, due to conformational constraints specific to the clade or diverse selection pressure forces specific to the population. This is supported by the fact that the class of bnAbs isolated from HIV-1 infected individuals is influenced by both the clade of the strain infecting each individual (the bnAbs-imprinting capacity of HIV-1 envelopes [82]) and by the ethnic origin [29].

It seems also that the magnitude of increase in resistance to bnAbs is variable depending on the strains, subtypes, and bnAbs. For subtype B viruses, between the two periods of 1988–1991 and 2007–2010, the mean IC_50_ increased four-fold for PG9 (V1V2g), six-fold for PGT121 (V3g), four-fold for VRC01 (CD4bs), and nine-fold for 3BNC117 (CD4bs), but did not for 10-1074 (V3g) [62]. While the mean IC_50_ remained below 1 μg/mL for PGT121 and 3BNC117, a reasonably achievable serum concentration by bnAbs infusion, it is important to note that a greater number of viruses had IC_50_ values > 10 μg/mL, hence were highly resistant to these bnAbs. This was due to single or combined point mutations in the HIV-1 Env, such as the loss of PNGS at position 332 for V3g bnAbs [73,83,84,85]. bnAbs are promising agents for prevention and treatment of HIV-1 infection, as shown by many studies in animal models, and more recently in human clinical trials [4,6,86,87,88,89,90,91]. In the future, the drift of HIV-1 towards an increasing resistance to NAbs could compromise the use of bnAbs in prophylaxis or treatment. It is interesting to notice that this rising resistance has happened while these drugs have not yet been largely utilized in the human population, illustrating once again the ability of HIV-1 to be one step ahead. In addition, resistant variants rapidly emerged in HIV-infected individuals during clinical trials using monotherapies of bnAbs [87,88,89]. Therefore, targeting several sites of vulnerability by using multiple bnAbs or multispecific antibodies enhances breadth and potency [62,90,92,93], and could address this problem. Achieving such multi-epitope protection with a vaccine will probably be necessary but challenging. 

The increasing resistance to bnAbs observed on T/F viruses sampled from infected individuals over the course of the epidemic raises the issue of the panels used to evaluate bnAbs and vaccine-induced anti-HIV-1 humoral response [63,94] and the necessity of updating them in the future. In other words, it is important to keep monitoring the sensitivity of HIV-1 to bnAbs over the course of the epidemic to better characterize its evolution for different clades and how it could impact immunization strategies.

## 5. A Future Vaccine to Overcome HIV-1 Diversity?

Over the last several years, many vaccine strategies have been explored to elicit broad neutralizing responses, based on the exceptional and promising capacity of bnAbs to prevent HIV-1 infection. These strategies aim to overcome two major roadblocks to successfully induce bnAbs: Engaging the correct precursor B cell of such antibodies and guiding their steps towards breadth, which requires very specific and unusual features such as extensive somatic hypermutation or CDR H3 length [95,96]. To date, two main complementary visions of immunogen designs have been inspired by the bnAbs characteristics. First, the lineage-based design is aimed at the activation of precursor B cells followed by sequential maturation of intermediate B cells to recapitulate the lineage of a bnAb. The second approach is to engineer epitope-based immunogens based on a tight knowledge of the sites of vulnerability [96,97,98]. If one of these approaches succeeds in the future—an already complicated task—it appears crucial to focus the effort on inducing antibodies which have the characteristics of the recent near-pangenotypic bnAbs N6 and VRC07-523, in order to tackle Env diversity [99,100]. However, as previously noticed, the induction of a single bnAb specificity is unlikely to provide full protection at the population level and could lead to the emergence of escape variants. Therefore, the design of a combined vaccine inducing two or more bnAb lineages with complementary neutralizing spectra would be mandatory. Several examples of HIV-1 infected individuals who developed NAbs targeting distinct sites of vulnerability have been described [101,102], showing that this is naturally possible. 

Among the various vaccine strategies explored, several are not based on mimicking the pathways of bNAbs but could have the potential to overcome HIV-1 diversity. One of these strategies is to use mosaic antigens covering the genetic diversity of circulating viruses worldwide by computational optimization of immunogens to maximize the coverage of potential epitopes [103]. Although low levels of NAbs were detected in several studies using mosaic Env antigens, some protection against simian/human immunodeficiency virus (SHIV) was observed in rhesus monkeys, associated with elicitation of Env-binding antibodies with enhanced effector functions [55,104,105,106]. Further evaluation of the breadth of protection conferred by this strategy is needed. Another approach is to use consensus Env immunogens designed to surrogate the HIV-1 sequences of the circulating viruses. Several studies have used a group M consensus Env to induce antibodies able to neutralize T/F viruses [107,108]. A heterologous response has been detected in these studies, suggesting that consensus Env might deserve to be further explored to overcome the Env diversity. 

## 6. Conclusions

HIV-1 diversity, which is particularly considerable for the envelope glycoproteins, is a major hurdle to achieving the goal of an effective vaccine. The interplay between HIV-1 evolution and the development the neutralizing response to HIV-1 infection has been explored in depth over the last few decades. The identification of bnAbs and sites of vulnerability of HIV-1 Env trimers, followed by comprehensive studies regarding the genesis of such bnAbs among the B cell repertoire, has raised the hope of generating a broad neutralizing response with innovative vaccine strategies. A range of new immunogens and vaccine design strategies are going to be tested in the next few years and lessons from immunogenicity trials will be crucial. So will be the results of ongoing clinical trials using bnAbs for HIV prevention and treatment, especially regarding their efficiency in different locations and against different genotypes. Considering the evolving nature of HIV-1 towards greater diversity and resistance to neutralization, as described in this review, new trials using combinations of bnAbs or near-pangenotypic antibodies will be needed to validate the full coverage of any immunization strategy, and to gain insight into the antibody concentrations required for effective protection. The fine-tuning required between breadth and serum concentration of NAbs in order to achieve protection remains unknown. Finally, the vaccine field would probably benefit from a prospective sampling of HIV-1 strains circulating around the globe to monitor their sensitivity to present and future active and passive immunization strategies.

## Figures and Tables

**Figure 1 vaccines-07-00074-f001:**
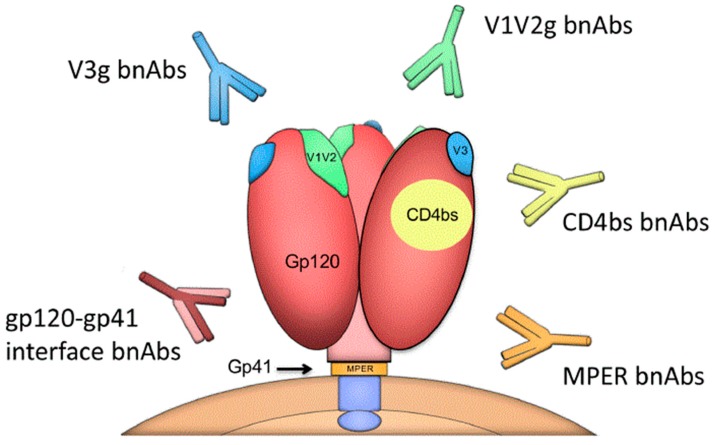
Sites of vulnerability on the HIV-1 envelope (Env) trimer and broadly neutralizing antibodies (bnAb) classes.

**Figure 2 vaccines-07-00074-f002:**
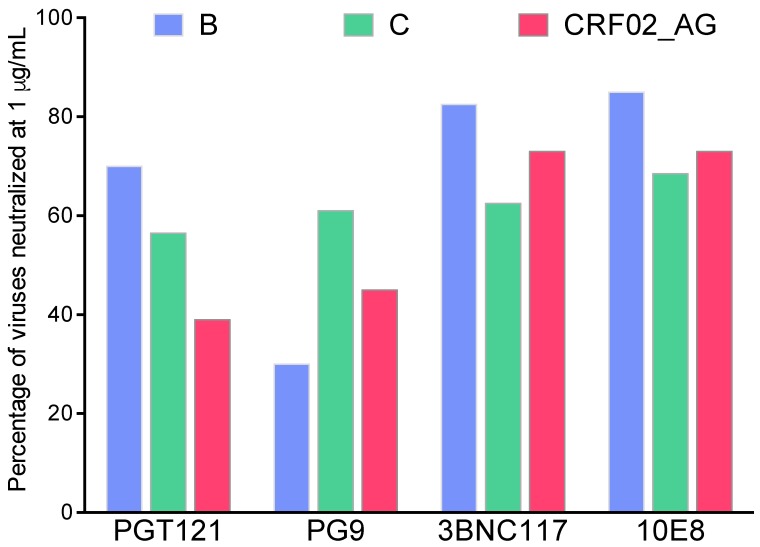
HIV-1 genetic diversity impacts its susceptibility to bnAbs. As an illustration, neutralization data were gathered for four bnAbs (PGT121, PG9, 3BNC117, and 10E8), each targeting a different site of vulnerability (V3g, V1V2g, CD4bs, and MPER, respectively), tested against three panels of transmitted-founder (T/F) viruses representing clade C (*n* = 200) [60], B (*n* = 40) [62], and CRF02_AG (*n* = 33) [61]. Based on individual half-maximal inhibitory concentration (IC_50_) values, the percentage of viruses neutralized at the bnAb concentration of 1 μg/mL within each clade is represented and highlights intersubtype discrepancies of the neutralization coverage.

**Figure 3 vaccines-07-00074-f003:**
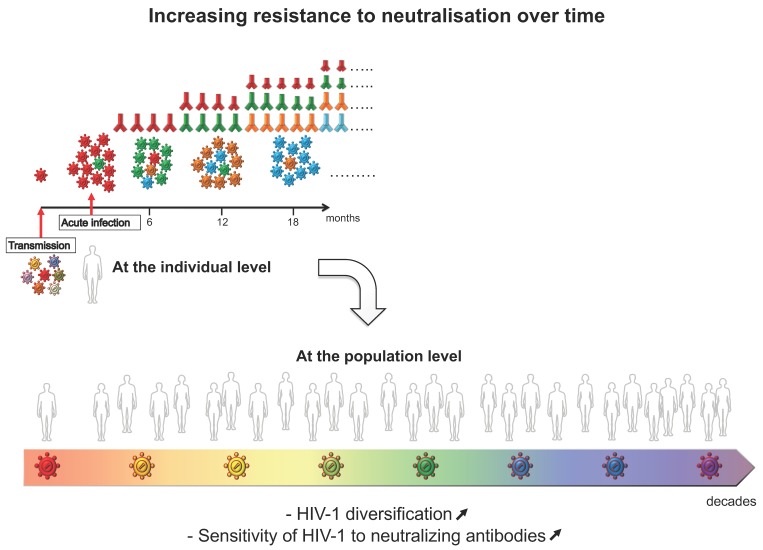
Schematic representation depicting the co-evolution between the virus and NAbs in an HIV-1 infected individual. (Top): Among a diverse population of variants from the transmitter, only a single variant (red) is sexually transmitted to the receiver in most of the cases. It gives rise to autologous NAbs (red) that exert a selective pressure. An escape variant becomes predominant (green), which induces, in turn, a specific autologous neutralizing response, and so on as the phenomenon continues. Progressively, this evolution at the individual level has a repercussion on the diversification of the HIV-1 species leading to an increasing resistance to NAbs at the population level (bottom).

**Figure 4 vaccines-07-00074-f004:**
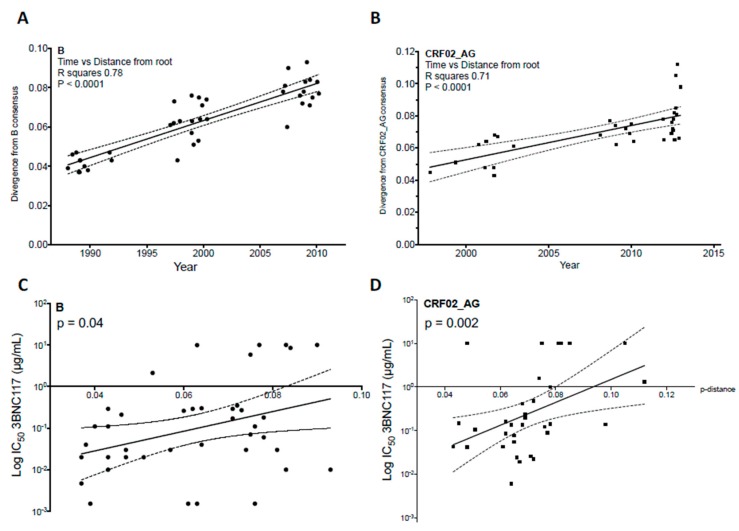
The diversification of HIV-1 over the course of the epidemic is associated with increased resistance to bnAbs. Panels (**A**–**D**) illustrate this phenomenon for two subtypes, by studying T/F viruses sampled shortly after seroconversion in HIV-1 infected individuals from 1987 to 2012 in France [61,62,73]. (**A**,**B**) The genetic distance of subtype B (**A**) and CRF02_AG (**B**) T/F variants from a consensus B or CRF02_AG, respectively, correlates with calendar year and increases over time. Sequences were codon-aligned with CRF02_AG consensus sequence (Los Alamos database). After exclusion of hypervariable regions, evolutionary divergence was estimated by distance matrix using MEGA software. Pearson correlation. (**C**,**D**) IC_50_ values of 3BNC117 were positively correlated with distance, indicating an increase in resistance. Spearman correlation.

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
