# Peer review of "Impact of HIV-1 Diversity on Its Sensitivity to Neutralization"

_vaccines, 2019, doi:10.3390/vaccines7030074_

Round 1

Reviewer 1 Report

Although progress towards an effective HIV vaccine has been glacial over the last 4 decades, the identification and characterization of broadly-neutralizing antibodies has provided arguably the first real prospect of progress towards this goal in the medium-term future. There has been intense interest in this area, and the present work competently reviews the field, and in particular the important issues around the evolution of resistance to particular idiotypes in the epidemic, and importantly, the contrasting picture emerging from different clades of group M viruses in respect of bNAb evolution. The work could be enhanced by a more rigorous analysis of the phylodynamics. 

The review is less strong - and arguably too sanguine - about the ways in which we can use this information to generate effective and evolution-resistant bNab-eliciting antigens, given the many fruitless efforts there have been in this direction.

In addition to some minor copy-editing for style, several of the figures are poorly presented and hard to read ( see especially Figure 2) and should be revised

Author Response

1. The review is less strong - and arguably too sanguine - about the ways in which we can use this information to generate effective and evolution-resistant bNab-eliciting antigens, given the many fruitless efforts there have been in this direction.

We agree with the fact that the results obtained with various vaccine strategies to date are not convincing. As summarized in this review (line 320-342), there are several roadblocks toward an effective vaccine regarding the immunogenicity and the protective efficacy of vaccine candidates. Several excellent reviews (ref 98,99; van Schooten and van Gils Retrovirology (2018)15:74) have already cover the subject and the possible strategies to address several of these roadblocks (and probably several new reviews in this special issue). Therefore, our aim was to focus on the Env diversity. We proposed several options using the aforementioned strategies currently tested to tackle the specific problem of Env diversity (combining immunogens to target multiple epitopes, using the broadest bnAbs recently discovered to design immunogens, line 342-351). In addition, we cited some studies that sought to counter this diversity using other strategies (line 352-371), but the results are too preliminary to judge on the their chance of success regarding the problem of Env diversity. We modified line 342-344 in order to contrast our point of view and appear less sanguine (“If one of these approaches succeeds in the future - an already complicated task - it appears crucial to focus the effort on inducing antibodies having the characteristics of the recent near-pan-genotypic bnAbs N6 and VRC07-523”).

2. In addition to some minor copy-editing for style, several of the figures are poorly presented and hard to read (see especially Figure 2) and should be revised.

Figure 2 has been revised for ease of reading, as requested both by reviewer 1 and 2 (point 4)

Reviewer 2 Report

This is a succinct review of the role of HIV-1 diversity on the sensitivity of HIV to antibody neutralization which references the work of the authors and many others in overlapping expertise areas. The authors evaluate various bnAbs and how effective they are globally and how this can change over time at the population level, with implications for vaccine design and antibody therapies. Overall, it's a nice contribution to the literature and is well written. Some comments are below: 

lines 73 and 82: "bNAbs after 2 to 3 years", "a very small fractions of patients, elite neutralizers develop bnAbs". It's commonly reported that about 20-30% of donors develop bnAbs, the elite neutralizers are among these and may be just more potent or broad. Perhaps this needs to be described more clearly.

Broadly neutralizing antibodies should be abbreviated as bnAbs upon first use and thereafter. 'Broadly neutralizing antibodies 'is spelled out on line 73, but could be abbreviated at line 22. Similar for NAbs and other acronyms. 

The text could be slightly improved/simplified in places for ease of reading. For example, line 73 "a few number of HIV-1 infected..." could be "some HIV-1 infected". 

Fig 2 could be redone to expand label text/reorient for better clarity. The text around lines 105-113 linked to figure 2 refers to clades A (line 108), B and C but the figure includes CRF AG instead of clade A which would better link up text and figure if that is possible. It might be useful also to include some comparison regarding how the TF virus neutralization profiles differ from other non-TF along with a mention of any features they have (e.g. fewer glycans etc). Also, in evaluating the activities of bnAbs against different clades, the donor origin of the bnAb may be a relevant point to make. Thus, if V1V2g bnAbs are less potent against subtype B, could this in part be because they were derived from non-B subtypes? Also, is the lower potency to clade B true for all V1V2g bnAbs, considering that this set of exhibit diverse binding mechanisms? Similarly for interface bnAbs which also have diverse binding mechanisms. For CD4bs bnAbs, is increased subtype A sensitivity specific to VRC01 class or does it cover VH1-46 class and b12 class also? And is this sensitivity linked to glycan differences in clade? 

The tendency for greater bnAb resistance at the population over decades is an interesting point. It makes me wonder how the timepoint from which bnAbs were accessed from the donor might influence the outcome. For example, a CD4bs bnAb isolated in 2009 may be more effective on viruses from a decade prior, but not against contemporaneous viruses. Perhaps one isolated from 2019 would fare better against a 2009 virus set as it may have adapted to population level sequence changes, so that it is better adapted than a 2009 bnAb. Thus, while increased population resistance occurs due to the increasing prevalence of escape variants in the population, these escape variants may create new targets for bnAbs that target the increasingly prevalent escape variants. This would tend to argue the idea that vaccines might be better constructed using the most recent Env sequences or predicted escape variants (a point the authors make). However, a lot of vaccine research focuses on Envs that have been around for decades. Maybe this is a point worth making/speculating more on.

Author Response

Reviewer 2

1. Lines 73 and 82: "bNAbs after 2 to 3 years", "a very small fractions of patients, elite neutralizers develop bnAbs". It's commonly reported that about 20-30% of donors develop bnAbs, the elite neutralizers are among these and may be just more potent or broad. Perhaps this needs to be described more clearly.

We have revised the manuscript. Line 80 :”20-30%” and line 103-104 :”Among individuals developing bnAbs, a small fraction of patients (1%) called elite neutralizers, develop very broad and highly potent bnAbs”

2. Broadly neutralizing antibodies should be abbreviated as bnAbs upon first use and thereafter. Broadly neutralizing antibodies 'is spelled out on line 73, but could be abbreviated at line 22. Similar for NAbs and other acronyms.

We agree with this comment. Abbreviations have been modified according to the recommendation (see changes in the text).

3. The text could be slightly improved/simplified in places for ease of reading. For example, line 73 "a few number of HIV-1 infected..." could be "some HIV-1 infected".

We have modified the text line 80. See point 1.

4. Fig 2 could be redone to expand label text/reorient for better clarity. The text around lines 105-113 linked to figure 2 refers to clades A (line 108), B and C but the figure includes CRF_02AG instead of clade A which would better link up text and figure if that is possible.

We have redone the Figure 2 to expand label text, focused the text on subtype B and C and clarified how this figure illustrates our point. Line 133-143.

5. It might be useful also to include some comparison regarding how the TF virus neutralization profiles differ from other non-TF along with a mention of any features they have (e.g. fewer glycans etc).

In this review, we have focused on T/F viruses as explained line 125-132, because it is more relevant for vaccine design. Regarding features of T/F, there are conflicting results that could confuse the reader in this review: several reports have suggested that subtypes A, C and D T/F viruses harbor Env with shorter variable loops and fewer potential N-linked glycosylation sites (PNGS) whereas these features were not confirmed in subtype B viruses (Ping et al. J Virol 2013 10.1128/JVI.03577-12, Derdeyn Science 2004 10.1126/science.1093137; Chohan J Virol 2005 10.1128/JVI.79.10.6528-6531.2005; Oberle Retrovirology 2016 10.1186/s12977-016-0299-0; Frost J Virol 2005 10.1128/JVI.79.10.6523-6527.2005 ; Wilen J Virol 2011 10.1128/JVI.00736-11)

6. Also, in evaluating the activities of bnAbs against different clades, the donor origin of the bnAb may be a relevant point to make. Thus, if V1V2g bnAbs are less potent against subtype B, could this in part be because they were derived from non-B subtypes?

This is a very interesting point indeed. We added a sentence to highlight this possibility (Line 145-148). 

7. Also, is the lower potency to clade B true for all V1V2g bnAbs, considering that this set of exhibit diverse binding mechanisms? Similarly for interface bnAbs which also have diverse binding mechanisms. For CD4bs bnAbs, is increased subtype A sensitivity specific to VRC01 class or does it cover VH1-46 class and b12 class also? And is this sensitivity linked to glycan differences in clade?

We agree with the reviewer that some differences could be observed within bnAb families although it’s not the case for all families. For example, within the CD4bs class, subtype A sensitivity is similar for VRC01 class and VH1-46 class (see Bricault et al., Cell Host Microbe, 2019) while the potency among the V1V2g bnAbs differ against subtype B. Rather than going into details for each class and subclass of bnAbs, we think that it’s preferable to stick to our point on HIV diversity and to limit our discussion to the bnAb classes as a whole.

8. The tendency for greater bnAb resistance at the population over decades is an interesting point. It makes me wonder how the timepoint from which bnAbs were accessed from the donor might influence the outcome. For example, a CD4bs bnAb isolated in 2009 may be more effective on viruses from a decade prior, but not against contemporaneous viruses. Perhaps one isolated from 2019 would fare better against a 2009 virus set as it may have adapted to population level sequence changes, so that it is better adapted than a 2009 bnAb. Thus, while increased population resistance occurs due to the increasing prevalence of escape variants in the population, these escape variants may create new targets for bnAbs that target the increasingly prevalent escape variants. This would tend to argue the idea that vaccines might be better constructed using the most recent Env sequences or predicted escape variants (a point the authors make). However, a lot of vaccine research focuses on Envs that have been around for decades. Maybe this is a point worth making/speculating more on.

We fully agree with the reviewer. As stated by the reviewer, we discussed about the importance to keep the surveillance of contemporaneous HIV T/F variants, update panels of virus used to assess the vaccine-elicited neutralizing responses and focus on these variants for the development of future vaccines. Lines 311-316).